# Prevalence and Risk Factors of Elevated Blood Pressure and Elevated Blood Glucose among Residents of Kajiado County, Kenya: A Population-Based Cross-Sectional Survey

**DOI:** 10.3390/ijerph17196957

**Published:** 2020-09-23

**Authors:** Anita Nyaboke Ongosi, Calistus Wilunda, Patou Masika Musumari, Teeranee Techasrivichien, Chia-Wen Wang, Masako Ono-Kihara, Charlotte Serrem, Masahiro Kihara, Takeo Nakayama

**Affiliations:** 1Department of Health Informatics, Kyoto University School of Public Health, Yoshida Konoe-cho, Sakyo-Ku, Kyoto 606-8501, Japan; nakayama.takeo.4a@kyoto-u.ac.jp; 2African Population and Health Research Centre, Manga Close, Nairobi P.O. Box 10787-00100, Kenya; calistuswilunda@yahoo.co.uk; 3Interdisciplinary Unit for Global Health, Centre for the Promotion of Interdisciplinary Education and Research, Kyoto University, Yoshida hon-machi, Sakyo-ku, Kyoto 606-8501, Japan; patoumus@yahoo.fr (P.M.M.); techasrivichien.teeranee.2a@kyoto-u.ac.jp (T.T.); kihara.masako.0612@gmail.com (M.O.-K.); kihara.masahiro.44z@st.kyoto-u.ac.jp (M.K.); 4International Institute of Socio-Epidemiology, Kitagosho-cho, Sakyo-ku, Kyoto 606-8336, Japan; 5Population Health Research Centre, College of Public Health, National Taiwan University, Taipei 100, Taiwan; am10312002@gmail.com; 6Department of Consumer Science, School of Agriculture and Biotechnology, University of Eldoret, Eldoret P.O. Box 1125-30100, Kenya; charlottejes@gmail.com

**Keywords:** prevalence, risk, pre-hypertension, hypertension, pre-diabetes, diabetes, sex, gender, elevated, Kajiado

## Abstract

Kenya is experiencing a rising burden of non-communicable diseases (NCDs), yet data to inform effective interventions are limited. We investigated the prevalence of elevated blood pressure, elevated blood glucose and their determinants in a rapidly urbanizing area in Kenya. Data on socio-demographics, dietary and behavioural risk factors, anthropometric measurements, blood pressure, blood glucose, plasma lipids and urinary biomarkers were collected from 221 men and 372 women (25–64 years). Multivariable logistic regression models assessed correlates of elevated blood pressure (EBP) and elevated blood glucose (EBG). Participants’ mean age was 38.0. ± 11.1 years. The prevalence rates of pre-hypertension and hypertension were 49.0% and 31.6% in men and 43.7% and 20.1% in women, respectively, while those of pre-diabetes and diabetes were 8.4% and 8.0% in men and 11.6% and 7.4% in women, respectively. The prevalence of Body Mass Index (BMI) ≥ 25 kg/m^2^ was higher in women (60.2%) than in men (39.7%). However, both the risk of EBP and EBG were stronger among men than among women. The high prevalence rates of EBP, EBG and overweight/obesity coupled with low physical activity and low fruit and vegetable intake predispose this population to a higher NCD risk. Interventions to mitigate this risk considering the sex differences are urgently required.

## 1. Introduction

Sub-Saharan Africa (SSA) is experiencing a growing burden of non-communicable diseases (NCDs), which has been attributed mainly to economic transition, rapid urbanisation, demographic transition and lifestyle changes [1,2]. Along with the longstanding challenges from infectious diseases, cardiovascular diseases (CVDs) and diabetes are among the NCDs rising rapidly in the SSA region [3]. It is projected that by 2030, NCD deaths will exceed the total deaths from communicable, maternal, perinatal and nutritional diseases [4,5].

Kenya, like other SSA countries, is faced with an unprecedented increase in the burden of NCDs, especially cancers and CVDs [6,7]. The prevalence of hypertension, one of the major risk factors of CVDs, has increased over the last 20 years in Kenya [8] with varying levels being reported in several communities. The prevalence of hypertension in adults was 21.4% in the adults of rural Kenya [9], 12.6% in Garissa County [10], 18% in Korogocho slums in Nairobi [11] and 50% in a population over 50 years old in Nakuru [12]. On the other hand, the prevalence of diabetes, another major risk factor of CVDs, has been estimated to be 2.0% among adults in Kenya according to the International Diabetes Federation 2017 report, amounting to almost 458,900 persons [13]. Future projections suggest that if current policy directions and interventions that give more emphasis on communicable diseases are sustained, deaths from communicable diseases may reduce by 48%, but deaths from NCDs will increase by 55% by 2030 [14].

Urbanization has emerged as a key driver of NCDs in many low- and middle-income countries (LMICs) and has been found to be associated with a higher prevalence of type 2 diabetes and higher levels of blood pressure and cholesterol [15]. In Kenya, 22.3% of the population is urban, with an annual urbanisation growth rate of 4.2%, which is almost double the national population growth rate of 2.4% [16]. Therefore, to tackle NCDs epidemic in Kenya, it is important to investigate the situation of NCDs risk factors, focusing on rapidly urbanizing areas. However, a majority of the NCD studies in Kenya have covered only poor urban (slums) or rural areas, lacking the information from regions in transition. Moreover, previous studies on NCDs in Kenya have mainly relied on self-reported estimates of risk factors, lacking the information on biological factors. Thus, we aimed to determine the prevalence of elevated blood pressure, elevated blood glucose and their determinants in Kajiado county, a typical rapidly urbanising area bordering the capital of Kenya. Kajiado County lies on soda volcanic bedrock resulting in salty water for domestic use, which may potentially expose the population to a greater risk of high salt consumption.

Kajiado County, historically home to the Maasai people, is now highly cosmopolitan with almost every ethnic community in Kenya represented in the major urban areas. The county has a total urban population of 395,051 representing 35% of the total population [17]. Kajiado is one of the four counties within the Nairobi Metropolitan area, making it a popular location for expanding settlements, industries and businesses. The county boasts of several major manufacturing factories and natural resources extractors, large-scale floriculture and horticulture, a vibrant real estate sector and thousands of small and medium sized enterprises absorbing at least 36% of the population [17].

Despite the positive contribution of urbanisation to the socio-economic activities, studies on the impact of urbanization on the livelihoods of the Maasai have found that unplanned land fragmentation and sale, the spread of human and industrial settlements, and the shrinking pastureland have implications for local communities whose major economic activity is pastoralism. This has further been compounded by environmental pollution [18], human-wildlife conflict [19,20] and mushrooming of informal settlements [16]. The urban population is projected to increase to 439,700 in 2020 and 489,399 in 2022 [17].

## 2. Materials and Methods

### 2.1. Study Setting

This study was conducted from November 2016 to August 2017 in Kajiado County, located in the Rift Valley and bordered by Nairobi, the capital of Kenya, to the north. Kajiado is a large county with a land area of 21,292.7 km^2^. It is home to 1,117,840 inhabitants [21]. Kajiado County is primarily semi-arid with a diverse topography ranging from volcanic hills and valleys to expansive plains, which posed a great challenge in transportation during field visits. The Maasai people, popularly known for their culture and tribal dress, make up the majority of the rural residents. Due to their semi-nomadic lifestyle and the natural features of Kajiado such as the Amboseli National Park and Nyiri desert, the Maasai are sparsely distributed in the study area. We used a caravan of strong vehicles to cross seasonal rivers and forests with no proper road network to reach remote settlements.

### 2.2. Study Population

Male and female residents aged 25 to 64 years, who were Kenyan citizens and had been residing in the county for at least 6 months and granted informed consent were included in the study. Pregnant women and women who had given birth in the last 6 months were excluded because pregnancy may modulate dietary habits, lifestyle, body weight and biomarkers.

### 2.3. Study Design and Sampling

The study is a population-based cross-sectional survey. Participants were selected using a stratified multi-stage cluster sampling method. First, enumeration areas (EAs) were stratified into urban and rural regions (EAs, 934 rural and 1021 urban) and within each stratum, 15 EAs were selected using the probability proportional to size (PPS) method, with the number of households in each EA as the measure of size. Second, from each selected EA, 25 households were selected by systematic sampling, and the Kish method was used to select one eligible participant from each selected household [22,23,24].

The sample size was calculated for both descriptive and analytical purposes using elevated blood pressure (pre-hypertension and hypertension) as the main outcome. For the descriptive purpose, assuming 95% confidence level, 5% margin of error (e^2^) and the prevalence of elevated blood pressure of 70% [25], the sample size was estimated to be 323 and inflated to 485 considering a design effect of 1.5.

For the analytical purpose, assuming that prevalence of elevated blood pressure is 80% and 70% in the population with and without unhealthy diet, respectively [8], α and β values of 0.05 and 0.2, respectively, the sample was estimated to be 626. The larger of the two sample sizes was selected for the study to ensure it was enough for the two purposes. The final sample size was decided to be 750 considering a possible response rate of 90% and the planned multivariable analysis.

### 2.4. Questionnaire

Data were collected using a structured questionnaire modelled along the WHO Stepwise survey questionnaire and based on the previous studies in Kenya [26,27] and other African countries [6]. The questionnaire was prepared in both English and Kiswahili, the two national languages in Kenya. In addition, two interviewers who were native Maasai language speakers interviewed the participants who understood only Maasai language (*n* = 11). The questionnaire was piloted in October 2016 to ensure face validity and test–retest reliability with one-week interval among 30 residents of Kajiado, who were not included in the main study. Kappa coefficients were calculated for categorical variables and intra-class correlation coefficients for continuous variables. The Kappa coefficients ranged between 0.44 and 1.00 and intra-class coefficients were 0.99–1.00, implying a moderate to excellent reliability of the questionnaire [28]. The variables in the questionnaire were broadly grouped as socio-demographic variables, behavioural risk factors, dietary risk factors, history of measurement of blood pressure and blood glucose and the diagnosis for hypertension and diabetes.

(1)Socio-demographic variables were age, sex, ethnicity, place of residence, length of stay, marital status, education, occupation and household size (9 items).(2)The behavioural risk factors included variables such as ever measured blood pressure, ever measured blood glucose, tobacco smoking, alcohol use, physical activity and concern about developing hypertension or diabetes (7 items). Physical activity was assessed by the amount of time and number of days one engaged in work-related, transport-related or sports activity that raised the breathing or heartbeat for more than 10 min [29]. Physical activity was classified as adequate if the participant engaged in the activity for at least 30 min and for a minimum of three days a week [30].(3)Dietary risk factors included variables such as number of meals eaten at home, place where meals are eaten outside home, number of times fruits and vegetables were consumed, use of cooking fat or oil; sugar or sugary foods and salt intake (9 items).

Additional questions included whether they had previously measured blood pressure and/or blood glucose, diagnosis for hypertension and diabetes, and the concern, knowledge, attitude and perception about hypertension, diabetes and body weight (13 items).

### 2.5. Anthropometric and Biological Measurements

Body weight was measured to the nearest 0.1 kg using a portable weighing scale (Omron HBF-223-G, OMRON Corporation, Kyoto, Japan) with participants dressed in lighter cloth and without shoes. Height, waist circumference and hip circumference were measured to the nearest 0.1 cm using a measuring tape [31]. Waist circumference was measured over a layer of light clothing. Participants were asked to stand with their feet close together making sure their weight was equally distributed on each leg and were breathing normally. Standing to the side of the participant, the last palpable rib and the top of the hip bone were located. The measuring tape was wrapped around the body in a horizontal position at the midpoint of the last palpable rib and the top of the hip bone making sure the tape was snug but not tight enough to cause compression of the skin. Hip circumference was measured as the maximal circumference over the buttocks [32]. Body mass index (BMI, kg/m^2^) was calculated as body weight (kg) divided by height in metre squared (m^2^).

Blood pressure was measured using a digital automatic blood pressure monitor (Omron HEM-7130-HP OMRON Corporation, Kyoto, Japan). Three blood pressure measurements were taken at 3-min intervals in a seated position after 15 min of rest, and the mean value of the last two readings was used for analysis. Systolic blood pressure (SBP) ≥ 140 mmHg or diastolic blood pressure (DBP) ≥ 90 mmHg or being on antihypertensive medication was classified as hypertension, and SBP 120–139 mmHg or DBP 80–89 mmHg was classified as pre-hypertension. SBP < 120 mmHg and DBP < 80 mmHg was classified as normal blood pressure.

Biological tests were conducted among participants who had fasted. For those who had eaten anything after 8 pm in the previous night, their appointments were rescheduled. Haemoglobin A1c (HbA1c), total cholesterol (TC), low-density lipoprotein cholesterol (LDL-C), high-density lipoprotein cholesterol (HDL-C) and triglycerides (TG) were measured using finger prick blood samples obtained using medical finger-stick devices (Cobas ^®^ b 101, Roche Diagnostics K.K., Tokyo, Japan). HbA1c readings of below 6.0% were considered normal; 6.0–6.4% was considered pre-diabetes, and 6.5% and higher or being on medication for diabetes was considered a diabetic state [33]. Abnormal concentrations of blood lipids were defined as follows: TC ≥ 240 mg/dL, LDL-C ≥ 130 mg/dL, HDL-C < 40 mg/dL, or TG ≥ 150 mg/dL [34].

Spot urine samples were collected from participants, and the sodium to potassium (Na/K) ratio was measured using a handy-sized urinary sodium-potassium ratio monitor (HEU-001F, Omron Corporation, Kyoto, Japan) and urinary sugar tested using a test strip. Although there is no generally accepted guideline for the urinary Na/K ratio, the cut-off points of between 1.0 and 2.0 exhibited the lowest CVD risk [35]. The cut off 1.0 is generally recommended as a target level while the cut-off point of 2.0 as a sub-optimal threshold. For the purpose of this study, the cut-off point of urinary Na/K ratio was set at 1.0 [36].

### 2.6. Data Collection

Data were collected by trained research assistants. To ensure high quality data, the research assistants were recruited based on completion of tertiary education and prior field survey experience. The research assistants underwent a two-day intensive training session to understand the objectives, methods and interview procedures and were involved in the pilot study to gain practical experience. Two of the research assistants had training in laboratory and biomedical science and were involved in collecting biological samples. Participants were interviewed in their homes and they were informed and reminded of the anthropometric and biological measurements the next day at a specified venue in the community.

In total, 750 participants were interviewed. During the study, 12 participants who declined to be interviewed were replaced, each of them by the next participant during systematic sampling. The common reason for declining the interview was lack of time due to other commitments.

### 2.7. Study Outcomes

Elevated blood pressure (pre-hypertension and hypertension) and elevated blood glucose (pre-diabetes and diabetes) were analysed as outcomes in this study to include all the participants who were at risk of developing CVDs, considering that pre-hypertension and pre-diabetes are not mild states but rather dangerous pre-clinical precursors [37].

A meta-analysis by Huang et al. (2013) indicated that pre-hypertension exists in up to 50 percent of people studied worldwide with devastating consequences. The condition raised the risk of coronary heart disease by 50%, risk of CVDs by 55% and risk of stroke by 71% [38]. Even pre-hypertension in the “low” range, a systolic reading between 120–140 mmHg caused 46% higher rate of heart disease [39]. Pre-diabetes on the other hand has also been shown to increase the risk of developing type 2 diabetes, heart disease, and stroke [40].

### 2.8. Statistical Analysis

Data were entered into EpiData software version 3.1 using the double entry method, validated and exported to Stata 15 software (Stata Corp LLC, College Station, TX, USA for analysis. We calculated the weights as the inverse probability of selecting a participant conditional on sex and place of residence then adjusted our analyses for the complex survey design. We summarised participants’ socio-demographic characteristics, prevalence of the health conditions and prevalence of the NCD risk factors using weighted descriptive statistics. We did not weight the data in the bivariate and multivariable analyses.

We excluded 157 (20.9%) participants with missing data on any of the key variables (BMI, HbA1c and blood pressure readings). Participants excluded and those included had a similar distribution of background characteristics (Appendix A), and thus, the exclusion is unlikely to have biased the results. Missing data arose mainly because biological measurements were carried out in a fasting state in the following morning of the interview day. Participants who participated in the interview but did not turn up for the measurements were classified as having missing data.

We calculated age-standardised prevalence for elevated blood pressure and elevated blood glucose using the direct standardisation method, with the 2009 national census data as the standard population. We stratified the prevalence of anthropometric, behavioural, and dietary risk factors by sex. We used bivariate and multivariable logistic regression models to obtain crude and adjusted odds ratios (AOR) with 95% confidence intervals (CI) for the associations of potential risk factors with elevated blood pressure and elevated blood glucose. We used the missing indicator method to account for missing covariate data [41]. Variables with *p* < 0.1 in bivariate analysis (Appendix A) were mutually adjusted for in multivariable analysis.

We found significant interactions between sex and factors such as age, BMI and physical activity and thus stratified the results by sex. Indeed, previous studies show that women and men have different levels of exposure and vulnerability to NCD risk factors [42]. Women are significantly more likely to be obese than men, which leads to their increased vulnerability to NCDs [43]. Additionally, social customs related to NCD risk have gender disparity [44,45], and women and men manifest certain NCD symptoms and risks differently [46].

### 2.9. Ethical Approval

The study protocol was approved by the Kyoto University Graduate School of Medicine (Japan), Ethics Committee (Ref: R0798) and the Institutional Research and Ethics Committee at Moi University College of Health Sciences (Ref: IREC/2016/214) (Kenya). Research permission was obtained from The National Commission for Science, Technology and Innovation in Kenya (Ref: NACOSTI/P/17/72838/13328) and the Kajiado County Ministry of Health office. All participants provided a written informed consent to participate. Participants with hypertension or diabetes defined by the study measurements or any other clinical conditions were referred to nearby health facilities.

## 3. Results

### 3.1. General Characteristics

Table 1 shows the background characteristics of the study participants. More than 80% of subjects in both sexes were married and a majority were of the Maasai ethnic tribe (45.2% men and 40.6% women), who resided mainly in the rural areas, followed by the Kikuyu tribe (15.1% in men and 24.5% in women), who resided mainly in the urban areas. In total, about 38% of the participants were living in urban areas, 53.8% were self-employed and 36.7% had attended at least secondary education with a higher proportion in men (39.2%) than in women (33.9%). Slightly less than a half (48.4% in men and 44.9% in women) had never measured their blood pressure, and a majority (57.1 % in men and 64.2% in women) had never measured their blood glucose.

### 3.2. Prevalence of the Clinical and Biological Indicators

As shown in Table 2, the prevalence of both pre-hypertension and hypertension was higher in men (49.0% and 31.6%, respectively) than in women (43.7% and 20.1%, respectively), while that of pre-diabetes and diabetes were similar between the sexes (8.4% and 8.0%, respectively in men and 11.6% and 7.4% in women). In overall, the age-standardised prevalence of hypertension (95% CI) was 26.2 % (21.5, 31.4) while the age-standardised prevalence of diabetes (95% CI) was 7.7% (5.1, 11.6). The lipids profile measurement indicated that only a small percentage of the participants had high concentration of TC (4.6%) and low concentration of HDL-C (2.4%), while around 10–20% had high concentration of LDL-C and TG. However, there was no significant difference between the men and the women. The median urinary Na/K ratio was 3.02, with 70% of the participants having a urinary Na/K ratio above the recommended level of 2.0 [36] and only 6% having the ratio of 1.0, which is the optimum level.

### 3.3. Prevalence of NCD Risk Factors

As illustrated in Table 3, the prevalence of both overweight and obesity was higher in women (31.6% and 28.6%, respectively) than in men (26.4% and 13.3%, respectively). Higher waist–hip ratio was also more prevalent in women (59.2%) than in men (45.9%). Current smokers comprised of 9.1% of the participants, with a much higher proportion among men (15.6%) than among women (1.7%). Similarly, alcohol consumption was much higher among men (25.5%) than among women (3.9%). The prevalence of adequate physical activity was 39.5%; being higher in men (45.0%) than in women (33.2%). Daily intake of fruits and vegetables was low for both men and women; with around 85% of the participants not having adequate intake. Consumption of high sugary foods and drinks was relatively low, with less than 15% consuming them daily in both sexes. Around 37% of men and 25.6% of women reported using fat rather than oil for cooking.

### 3.4. Factors Associated with Elevated Blood Pressure and Elevated Blood Glucose

Table 4 shows the factors associated with elevated blood pressure in multivariable analysis (results of bivariate analysis are presented as Appendix A). Association of age with elevated blood pressure was dose-dependent in both sexes as AORs increased from 0.72 (0.31, 1.70) to 2.08 (0.56, 7.70) in men and from 1.81 (1.08, 3.04) to 4.89 (1.62, 14.79) in women for 35–54 and 55–64 years, respectively, which was, however, statistically significant only in women. Similarly, being of non-Maasai ethnicity was significantly associated with elevated blood pressure with AOR 2.62 (1.22, 5.62) for Kikuyu and 2.48 (1.18, 5.21) for others among only women. Self-employed men were significantly more likely to have elevated blood pressure compared with the unemployed men with AOR 3.72 (1.13, 12.22). BMI was associated with elevated blood pressure in a dose-dependent manner in both sexes but more strongly in men, with AORs of 3.23 and 17.13 in men and 1.47 and 2.30 in women for BMI 25–29 and BMI ≥ 30, respectively, with reference to BMI < 25.

As illustrated in Table 5, age was also significantly associated with elevated blood glucose in a dose-dependent manner in both sexes, with AOR increasing from 2.32 (0.82, 6.56) to 3.94 (1.15, 13.48) in men and from 1.20 to 3.96 in women for 35–54 years and 55–64 years, respectively, with reference to 25–34 years. BMI was also significantly associated with elevated blood glucose in a dose-dependent manner in both sexes but more strongly in men, with AOR increasing from 2.49 (0.89, 6.91) to 5.33 (1.78, 16.00) in men and from 0.74 (0.33, 1.68) to 2.43 (1.16, 5.06) in women of BMI 25–29 and BMI ≥ 30, respectively, with reference to BMI < 25. Higher waist–hip ratio was significantly associated with elevated blood glucose only in women (AOR = 3.18).

## 4. Discussion

We quantified the distribution of the levels of blood pressure and blood glucose and other NCD risk factors and investigated the correlates of elevated blood pressure and elevated blood glucose in Kajiado County. This is the first study in Kenya that investigates the urbanizing local community in terms of NCD risk factors, incorporating the measurement of blood pressure and blood glucose, plasma lipids and urinary Na/K ratio. Overall, we found that a substantial proportion of participants had elevated blood pressure or elevated blood glucose, which were associated with only a limited number of factors, including age and BMI. Moreover, although the prevalence of dyslipidaemia was still low, unhealthy behaviours, especially low physical activity and low vegetable/fruit intake were prevalent in both sexes, with smoking and drinking habits being prevalent only in men.

### 4.1. Prevalence of the Elevated Blood Pressure and Elevated Blood Glucose

The overall crude prevalence of elevated blood pressure was 72.7% (80.6% in men and 63.8% in women). The age-standardized prevalence of hypertension was 26.2%, which is slightly higher than the national average of 24.5% [47]. On the other hand, the crude prevalence of elevated blood glucose was 17.6% (16.4% in men and 19.0% in women). When standardized for age, the prevalence of diabetes of 7.7% was higher than the national average of 2.4% [48], suggesting that our study area had relatively higher levels of cardio-metabolic risk factors for NCDs.

Despite such high prevalence of elevated blood pressure and glucose, a worryingly high proportion (46.7%) of participants had never measured their blood pressure, and over half (60.4%) had never measured their blood glucose. Even worse, only 34.7% of participants having hypertension and 23.7% of those having diabetes were aware of their condition, meaning that most of the patients with elevated blood pressure and elevated blood glucose were left untreated. The national stepwise survey found a similar pattern whereby 56% and 88% of the respondents had never had their blood pressure and blood glucose measured, respectively [49]. Other subpopulation studies in Kenya have also reported on poor awareness, control and treatment for blood pressure and blood glucose [47]. In view of the increasing burden of hypertension and diabetes and the potentially devastating consequences of these health conditions, prevention programs are urgently needed to sensitize the population and screen individuals at risk of NCDs. Since it has been observed that rapid urbanisation is causing an upsurge in ischaemic heart disease and metabolic disorders [50], programs in rapid urbanizing areas such as our study area should be a priority.

### 4.2. Prevalence of other NCD Risk Factors

Overweight/obesity prevalence was high in both men (39.7%) and women (60.2%). Similar prevalence of overweight and obesity, with a sex disparity, have been reported consistently in Kenya and in many countries in Africa [51].

Expectedly, the proportion of alcohol consumers or tobacco smokers was higher (Table 3) in men (25.5% and 15.6%) than in women (3.9% and 1.7%). However, of greater concern is that half of current drinkers were younger than 35 years (results not shown), which is contrary to what was found in rural areas of western Kenya where drinking habit was more prevalent among older (45–54 years) men [52]. If this pattern is a consequence of urbanisation, it is possible that the prevalence of alcohol drinking would continue to increase over time, highlighting the need for prevention programs particularly targeting young people.

Although over 90% of our participants were involved in physical activity (work, walking, cycling), only 39% engaged in adequate physical activity. More men were physically active than women even though in Kenya physical activity has generally been viewed as a means to have work completed and is not linked to health [53]. In view of the previous studies in Kenya that showed sedentary lifestyles prevailing among Kenyan children and adolescents and the lack of national policy for promotion of physical activity in Kenya [54,55], interventions to promote physical activity in this study population are needed.

Our evaluation of quality of diet by examining four key elements (fruit and vegetable consumption, salt intake, intake of sugary foods and beverages and the use of cooking oil/fat) revealed a pattern similar to that reported in the national NCD survey [49]. Of importance among them would be the low fruit and vegetable intake. Although WHO recommends fruit and vegetable intake of at least five servings (400 g) daily for the prevention of chronic diseases [31], over half of the participants in our study population did not consume vegetables daily and over 70% did not consume fruits daily.

Although we are unable to compare our results on the complete profile of plasma lipids with those of other studies in Kenya, the proportion of subjects with high TC or low HDL-C concentrations in this study is lower than the national estimates [49]. About 5% of our study participants had high TC concentration compared to about 10% of adults nationally. In addition, whereas 50% of men and 60% of women had low HDL-C concentration at the national level, over 90% of the subjects in our study had optimal HDL-C concentration. The reason for this substantial difference is not clear and deserves further investigation.

We introduced measurement of urinary Na/K ratio out of the concern that salt intake by the inhabitants of our study area might be high given that the water used for domestic purposes in the area is salty due to the underlying soda volcanic rock. Our study revealed that the median of urinary Na/K ratio was only an average of international standard, but still 70% of the participants had urinary Na/K ratio above the recommended value of 2.0 [36]. This important finding could be considered in designing intervention strategies. According to the INTERSALT study—a large multicentre international collaborative study on salt and blood pressure—a reduction of urinary Na/K ratio from 3.09 to 1.00 reduces the population average of SBP by about 3.36 mmHg [56,57]. Although assessment of urinary Na/K ratio is convenient and simple, studies that have included this measurement remain remarkably scarce in Kenya. Therefore, it is important to consider the inclusion of urinary Na/K since it is key for future prevention programs. In addition, assessment of urinary Na/K ratio can overcome under-reporting of sodium intake, which is one of the major difficulties observed in dietary surveys [58].

### 4.3. Correlates of Elevated Blood Pressure and Elevated Blood Glucose

BMI was directly associated with both elevated blood pressure and elevated blood glucose in both sexes. The association of BMI with hypertension and diabetes has been previously reported in the literature from both developing and developed settings [6,59]. In Africa, an ecological study on the trends of obesity and diabetes from 1980 to 2014 found a strong positive association between mean BMI level and the prevalence of diabetes in both men and women [60], and other studies have shown associations between BMI and hypertension [61,62]. In Kenya, Olack et al. (2015) found that overweight and obesity were associated with hypertension with odds ratios of 1.7 and 2.0, respectively, among respondents living in the Kibera slum in Nairobi [27]. In view of the high prevalence of overweight and obesity in our study population and the report by the International Food Policy Research Institute (IFPRI) [63] that indicated a rapid rise in peoples’ BMI especially in urban Kenya, there is need to address the problem of overweight/obesity.

Association of BMI with elevated blood pressure and elevated blood glucose was stronger in men than in women in our study, which is consistent with previous studies in Kenya [25,47] and in other African countries [64]. This was because the proportion of overweight and obese participants with normal blood pressure or with normal blood glucose was much higher in women than in men, while their proportions among elevated blood pressure group or elevated blood glucose group do not differ as much between sexes (Appendix A).

Regarding the relationship between blood pressure and BMI, another interesting finding from our study is that 52% of participants with normal BMI had pre-hypertension (Appendix A). This finding is consistent with previous studies in Kenya [25] and developed countries [65], suggesting the mechanism of blood pressure elevation that is independent of increased body weight. Although the exact mechanism is unknown, it has been argued that it may be due to some environmental or genetic factors [66].

From these observations, it is suggested that, though a weight reduction approach is essential in the prevention and management of elevated blood pressure and elevated blood glucose, screening of hypertension should not solely focus on individuals with a high BMI (overweight and obese) but also on those with normal or lower BMI.

In our study, men and women aged 55–64 years had the highest prevalence rates of both elevated blood pressure and elevated blood glucose. In a recent study covering four different regions of Kenya, Ogola and others noted that the prevalence of pre-hypertension increased with age, peaking at 50–69 years (49.9%) followed by a decline. The decline coincides with an increase in hypertension, most likely reflecting a transition to hypertension [67]. In this view, there is urgent need to promote prevention, early detection and management of the chronic conditions to avoid further complications.

Ethnicity-wise, the Kikuyu and other ethnic groups were at higher odds of elevated blood pressure compared to the Maasai in women. Although the exact reasons of this ethnic difference are not clear from our study, it might be that, although the lifestyle of the residents of Kajiado is changing due to urbanisation, the Maasai women may still be keeping some of their traditional ways of life that are preventive of elevated blood pressure.

Finally, our study indicated potential association between self-employment and elevated blood pressure in men. Although some subpopulation studies in Kenya [68], Saudi Arabia [69] and Turkey [70] reported an association between employment status and dyslipidaemia, another risk factor for CVDs, other larger studies found no significant association between occupation and dyslipidaemia [71], arguing that this association would be a result of confounding by unknown factors.

Our study found several factors that could be targeted to improve the NCD situation and overall health in Kajiado County. First, the unhealthy dietary and lifestyle behaviours should be addressed. Most (90%) study participants were engaged in physical activity, but only 39.5% were having adequate physical activity. There is need for sensitization programs to increase awareness on the adequate level physical activity for desired health benefits. The Maasai people are known for their intense physical activity due to pastoralism. However, as we observed, the situation seems to be changing due to urbanisation, and sensitization programs will need to address this.

Secondly, the low intake of fruits and vegetables, which is partly due to the semi-arid nature of the climate in Kajiado, can be addressed by the county government through both incentives and measures that can ensure availability on the local markets. For example, the agricultural companies producing fruit and vegetables in green houses could be required to make a portion of them available to the local market. The sensitization programs mentioned above could also encourage consumption of the fruits and vegetables.

Thirdly, young men below 35 years were found to be the group mostly engaged in alcohol consumption in the county. Given that the population is projected to increase rapidly due to urbanisation, this habit especially among young people needs to be addressed urgently by the county government.

Fourthly, the high prevalence of overweight and obesity in both men and women could be addressed through community-based interventions to promote a healthy lifestyle and dietary behaviours such as increasing physical activity and the consumption foods low in energy and fat and high in fibre. Poor awareness of NCDs should also be addressed through community-based interventions, and screening, early diagnosis, treatment and rehabilitation services for hypertension and diabetes should be made available. The community should be sensitized to the risk of pre-hypertension even among those with normal BMI. The NCD control strategy in Kajiado needs to address the health challenges that are associated with rapid urbanisation and socio-economic development.

Given that Kajiado is located in a salty water zone and the observation that over 70% of the participants had a urinary Na/K ratio of over 2.0, there is need for further research on the effect of the salty water on the health of the people, practical ways to desalinate the water for domestic use and closer observation on the salt consumption in order to curb future health problems.

### 4.4. Strengths and Limitations

The study has the following strengths. Firstly, this is among the first NCD survey reports to cover an entire county. Secondly, unlike previous studies on NCDs in Kenya, which mainly relied on self-reported risk factor measurements, this study collected data on anthropometric and cardio metabolic markers together with self-reported data. The availability of biomarker measurements provides additional information that has largely been lacking. Thirdly, this study covered both rural and urban residents and both the natives (Maasai) and non-natives, which improves generalizability of the findings. As it was noted in the previous studies on hypertension and diabetes in Africa, it is essential for studies to capture both rural and urban populations to address disparities in risk factors [72]. Fourthly, analysing the data by sex, our study identified the sex disparity regarding the distribution of NCD risk factors and their associations with elevated blood pressure and elevated blood glucose, providing useful information for designing sex and gender-matched prevention programs in future.

The study has the following limitations. Firstly, data on fruit and vegetable consumption, cooking oil and cooking fat use, salt use, tobacco use, alcohol use and physical activity were self-reported and hence prone to information bias. These factors are known to be associated with elevated blood pressure or elevated blood glucose; however, we did not replicate such associations in this study probably partly due to this bias. Secondly, we excluded 20.9% of the participants in the data analysis due to missing information. Even though we observed no statistically significant difference between the participants included and those excluded from the analysis based on the variables in the questionnaire (Appendix A), the two groups might have differed by other unmeasured variables. Thirdly, although excessive sodium intake is an established risk factor for hypertension and diabetes [73,74], the question on salt intake was poorly answered, resulting in a lot of missing values, and was thus not used in the analysis. Fourthly, in clinical practice, a diagnosis of hypertension requires multiple measurements on several occasions, but we took measurements on a single day. However, this potential bias was minimized by using the mean of the last two of the three blood pressure measurements.

Finally, data from a cross-sectional study design prohibits causal interpretation. Our findings may have been affected by reverse causality, where participants with diabetes or hypertension prior to the study might have modified their diet and lifestyle. To mitigate this study design limitation, we recommend setting up large-scale cohort studies to investigate NCD risk factors in Kenya. Currently, such studies are based mainly in high income countries.

## 5. Conclusions

Our study revealed that over two-thirds of the participants had elevated blood pressure and about one fifth had elevated blood glucose. Overall, our results indicate that a substantial proportion of participants, in Kajiado County are at risk of developing NCDs. There is need for urgent introduction of intervention programs for NCDs, especially in the areas experiencing rapid lifestyle changes due to urbanisation. The intervention programs should be sensitive to sex differences in risk factors, such as the high prevalence of overweight and obesity among women with normal blood pressure and high prevalence of elevated blood pressure among men with normal BMI.

## Figures and Tables

**Table 1 ijerph-17-06957-t001:** Socio-demographic characteristics of participants segregated by sex among the residents of Kajiado County, Kenya. (*n* = 593).

Variables	Total		Men		Women	
	*n*	(%)	*n*	(%)	*n*	(%)
Unweighted	593		221		372	
Weighted	596		316		280	
Age (years)						
25–34	236	(39.6)	114	(36.1)	122	(43.6)
35–54	246	(41.2)	119	(37.7)	127	(45.2)
55–64	114	(19.1)	83	(26.2)	31	(11.2)
Marital status						
Single (Never married/ divorced/widowed)	97	(16.3)	54	(17.2)	43	(15.2)
Married	499	(83.8)	262	(82.8)	237	(84.8)
Ethnicity						
Maasai	257	(43.1)	143	(45.2)	114	(40.6)
Kikuyu	116	(19.5)	47	(15.1)	69	(24.5)
Other	223	(37.4)	126	(39.7)	97	(34.9)
Residence						
Rural	370	(62.1)	196	(62.2)	174	(62.1)
Urban	226	(37.9)	120	(37.8)	106	(37.9)
Education						
Basic	377	(63.3)	192	(60.8)	185	(66.1)
High school/higher	219	(36.7)	124	(39.2)	95	(33.9)
Occupation						
Unemployed	135	(22.6)	48	(15.1)	87	(31.0)
Employed	141	(23.7)	95	(30.1)	46	(16.4)
Self-employed	320	(53.8)	173	(54.8)	147	(52.6)
Ever measured blood pressure						
Yes	308	(51.6)	161	(50.8)	147	(52.5)
No	278	(46.7)	153	(48.4)	125	(44.9)
Missing data	10	(1.7)	3	(0.8)	7	(2.6)
Ever measured blood glucose						
Yes	229	(38.4)	134	(42.3)	95	(34.0)
No	360	(60.4)	180	(57.1)	180	(64.2)
Missing data	7	(1.2)	2	(0.7)	5	(1.7)

**Table 2 ijerph-17-06957-t002:** Prevalence of cardio-metabolic risk factors of non-communicable diseases (NCDs **) by sex among the residents of Kajiado County, Kenya (*n* = 593).

Variable	Total		Men		Women		
	*n*	% (95% CI)	*n*	% (95% CI)	*n*	% (95% CI)	*p*-Value *
Unweighted	593		221		372		
Weighted	596		316		280		
Blood pressure (mmHg)							<0.01
Normal (<120 and <80)	163	27.3 (21.9, 33.5)	61	19.4 (13.9, 26.4)	102	36.2 (29.2, 44.0)	
Pre-hypertension (120–139/80–89)	277	46.5 (41.9, 51.2)	155	49.0 (42.6, 55.5)	122	43.7 (38.4, 49.1)	
Hypertension (≥140 and ≥90)	156	26.2 (21.5, 31.4)	100	31.6 (24.0, 40.3)	56	20.1 (15.5, 25.5)	
HbA1c (%)							0.59
Normal (<6.0)	491	82.4 (76.5, 87.0)	264	83.6 (75.9, 89.1)	227	81.1 (73.8, 86.6)	
Pre-diabetes (6.0–6.4)	59	9.9 (7.0, 14.0)	27	8.4 (4.76, 14.7)	32	11.6 (8.2, 16.1)	
Diabetes (≥6.5)	46	7.7 (5.1, 11.6)	25	8.0 (4.6, 13.4)	21	7.4 (4.1, 13.0)	
Total cholesterol (mg/dl)							0.14
Optimal (≤239)	568	95.4 (92.4, 97.2)	297	93.9 (88.3, 96.2)	271	97.0 (94.2, 98.5)	
High (≥240)	28	4.6 (2.8, 7.6)	19	6.1 (3.1, 11.7)	9	3.0 (1.5, 5.8)	
Low density lipoprotein (mg/dL)							0.12
Optimal (≤129)	487	81.7 (77.0, 85.7)	254	80.4 (73.2, 86.1)	233	83.2 (78.0, 87.4)	
High (≥130)	82	13.7 (10.2, 18.3)	52	16.4 (11.1, 23.7)	30	10.7 (7.0, 16.0)	
Missing data	27	4.5 (2.9, 7.1)	10	3.2 (1.3, 7.3)	17	6.1 (3.6, 10.1)	
High density lipoprotein (mg/dL)							0.35
Low (<40)	15	2.4 (1.2, 5.1)	8	2.5 (0.8, 7.3)	7	2.3 (1.1,4.9)	
Optimal (≥40)	554	93.1 (89.2, 95.6)	298	94.4 (88.6, 97.3)	256	91.6 (86.4, 94.9)	
Missing data	27	4.5 (2.9, 7.1)	10	3.1 (1.3, 7.3)	17	6.1 (3.6, 10.1)	
Triglycerides							0.44
Optimal (≤199)	484	81.2 (75.5, 85.9)	262	82.9 (73.8, 89.3)	222	79.4 (73.2, 84.5)	
High (≥200)	112	18.8 (14.1, 24.5)	54	17.1 (10.7, 26.2)	58	20.6 (15.5, 26.8)	
Sodium-potassium ratio							0.22
Lower (<1.0)	41	6.7 (4.1, 10.9)	16	5.0 (2.1, 11.1)	25	8.7 (5.1, 14.5)	
Higher (>1.0)	555	93.3 (89.2, 95.9)	300	95.0 (88.9, 97.9)	255	91.3 (85.5, 95.0)	

CI: Confidence Interval. * *p*-values are from χ² tests adjusted for the study design comparing men and women. ** NCDs refer to hypertension and diabetes in this study. HbA1c: Haemoglobin A1c.

**Table 3 ijerph-17-06957-t003:** Anthropometric, behavioural and dietary risk factors for non-communicable diseases (NCDs **) by sex among the residents of Kajiado County, Kenya (*n* = 593).

Variable	Total		Men		Women		
	*n*	% (95% CI)	*n*	% (95% CI)	*n*	% (95% CI)	*p*-Value *
Unweighted	593		221		372		
Weighted	596		316		280		
Anthropometric							
Body mass index (kg/m^2^)							<0.01
Normal (<25.0)	302	50.7 (41.4, 59.9)	190	60.3 (47.8, 71.6)	111	39.8 (31.8, 48.4)	
Overweight (25.0–29.9)	172	28.8 (23.7, 34.6)	84	26.4 (18.3, 36.5)	89	31.6 (27.4, 36.2)	
Obese (30.0 and above)	122	20.5 (15.5, 26.7)	42	13.3 (8.8, 19.7)	80	28.6 (22.1, 36.1)	
Waist–hip ratio							0.04
Normal (<0.9/men and <0.85/women)	274	46.0 (37.6, 54.7)	164	51.9 (39.4, 64.1)	110	39.4 (32.3, 47.0)	
High (≥0.9/men and ≥0.85/women)	311	52.2 (43.7, 60.5)	145	45.9 (34.2, 58.1)	166	59.2 (51.5, 66.5)	
Missing data	11	1.8 (0.6, 5.7)	7	2.2 (0.6, 8.1)	4	1.4 (0.5, 4.0)	
Behavioural							
Smoking							<0.01
Not smoking	542	90.9 (8.6, 94.3)	267	84.4 (75.5, 90.5)	275	98.3 (94.5, 99.5)	
Currently smoking	54	9.1(5.7. 14.1)	49	15.6 (9.5, 24.5)	5	1.7 (0.5, 5.5)	
Alcohol consumption							<0.01
Not drinking	505	84.6 (77.9, 89.6)	236	74.5(64.2, 82.6)	269	96.1(91.9, 98.2)	
Currently drinking	91	15.4 (10.4, 22.1)	80	25.5 (17.4, 35.8)	11	3.9 (1.8, 8.1)	
Adequate physical activity							0.07
Yes	235	39.5 (33.0, 46.4)	142	45.0 (34.5, 56.0)	93	33.2 (26.8, 40.4)	
No	346	58.1 (51.1, 64.7)	166	52.6 (41.6, 63.4)	180	64.2 (56.9, 70.8)	
Missing data	15	2.5 (1.2, 5.1)	8	2.4 (0.9, 6.4)	7	2.6 (1.4, 4.7)	
Dietary							
Fruits and veg daily intake							0.51
Yes	91	15.3 (10.5, 21.6)	48	15.2 (9.3, 23.7)	43	15.3 (10.5, 21.9)	
No	501	84.1 (77.5, 88.9)	267	84.5 (76.0, 90.4)	234	83.7 (76.6, 88.6)	
Missing data	4	0.7 (0.3, 2.1)	1	0.3 (N/A)	3	1.0 (0.4, 3.9)	
High sugary foods and drinks							0.25
Daily	78	13.0 (9.2, 18.2)	38	12.0 (6.7, 20.8)	40	14.2 (9.9, 19.8)	
Weekly	216	36.2 (30.1, 42.8)	115	36.3 (27.3, 46.3)	101	36.2 (29.8, 43.0)	
Occasionally	295	49.4 (44.7, 54.2)	163	51.4 (44.4, 58.3)	132	47.2 (41.6, 52.8)	
Missing data	8	1.3 (0.6, 3.1)	1	0.3 (N/A)	7	2.5 (1.0, 6.0)	
Use cooking fat/oil							0.02
Mainly use cooking oil	395	66.4 (52.4, 77.9)	193	61.0 (44.9, 75.0)	202	72.4 (59.5, 82.4)	
Mainly use cooking fat	188	31.5 (19.9, 46.0)	116	36.7 (22.5, 53.7)	72	25.6 (15.8, 38.5)	
Missing data	13	2.2 (1.0, 4.7)	7	2.3 (0.9, 6.0)	6	2.0 (0.9, 4.6)	

CI: Confidence Interval. N/A: The number with missing data was very small; thus, the confidence interval could not be computed. * *p*-values are from χ² tests adjusted for the study design comparing men and women. ** NCDs refer to hypertension and diabetes in this study.

**Table 4 ijerph-17-06957-t004:** Results of multivariate logistic regression analysis with elevated blood pressure as outcome among the residents of Kajiado County, Kenya (*n* = 593).

Variable	Total (*n* = 593)		Men (*n* = 221)		Women (*n* = 372)	
	AOR (95% CI)	*p*-Value	AOR (95% CI)	*p*-Value	AOR (95% CI)	*p*-Value
Age (Years)						
25–34	1 (Ref.)		1 (Ref.)		1 (Ref.)	
35–54	1.36 (0.88–2.08)	0.16	0.72 (0.31–1.70)	0.46	1.81 (1.08–3.04)	0.03
55–64	3.54 (1.56–8.00)	<0.01	2.08 (0.56–7.70)	0.28	4.89 (1.62–14.79)	0.01
Sex						
Women	1 (Ref.)					
Men	2.37 (1.48–3.80)	<0.01	^a^	^a^	^a^	^a^
Ethnicity						
Maasai	1 (Ref.)		1 (Ref.)		1 (Ref.)	
Kikuyu	2.05 (1.09–3.82)	0.03	1.60 (0.45–5.62)	0.47	2.62 (1.22–5.62)	0.01
Other	1.92 (1.08–3.42)	0.03	1.59 (0.57–4.42)	0.38	2.48 (1.18–5.21)	0.02
Residence						
Rural	1 (Ref.)		1 (Ref.)		1 (Ref.)	
Urban	1.14 (0.68–1.91)	0.61	0.93 (0.36–2.39)	0.89	1.17 (0.62–2.22)	0.63
Occupation						
Unemployed	1 (Ref.)		1 (Ref.)		1 (Ref.)	
Employed	1.07 (0.60–1.91)	0.82	2.69 (0.78–9.27)	0.12	0.79 (0.39–1.59)	0.51
Self employed	1.72 (1.06–2.80)	0.03	3.72 (1.13–12.22)	0.03	1.57 (0.91–2.70)	0.10
Smoking						
Not smoking	1 (Ref.)		1 (Ref.)		1 (Ref.)	1 (Ref.)
Currently smoking	1.09 (0.42–2.82)	0.87	1.14 (0.38–3.44)	0.81	N/A	*
Alcohol consumption						
Not drinking	1 (Ref.)		1 (Ref.)		1 (Ref.)	
Currently drinking	1.47 (0.72–3.02)	0.29	2.33 (0.90–6.07)	0.08	1.08 (0.34–3.50)	0.89
BMI (Kg/m^2^)						
Normal (<25.0)	1 (Ref.)		1 (Ref.)		1 (Ref.)	
Overweight (25.0–29.9)	1.87 (1.20–2.94)	0.01	3.23 (1.20–8.68)	0.02	1.47 (0.86–2.51)	0.16
Obese (30.0 and above)	2.97 (1.68–5.24)	<0.01	17.13 (1.85–158.46)	0.01	2.30 (1.23–4.33)	0.01
Waist–hip ratio						
Normal (<0.9/men and <0.85/women)	1 (Ref.)		1 (Ref.)		1 (Ref.)	
High (≥0.9/men and ≥0.85/women)	1.11 (0.73–1.69)	0.62	1.31 (0.56–3.06)	0.54	1.01 (0.61–1.68)	0.95
Missing data	1.63 (0.31–8.56)	0.56	N/A	*	0.74 (0.11–4.81)	0.76
HbA1c (%)						
Normal (<6.0)	1 (Ref.)		1 (Ref.)		1 (Ref.)	
Pre-diabetes (6.0-6.4)	1.00 (0.52–1.95)	0.10	1.97 (0.40–9.76)	0.41	0.86 (0.40–1.84)	0.69
Diabetes (≥6.5)	2.37 (0.75–7.50)	0.14	0.86 (0.82–9.07)	0.90	2.56 (0.66–9.83)	0.17

CI: Confidence Interval; AOR: Adjusted Odds Ratio (all the variables in the table were mutually adjusted for each other in the multivariable analysis). Ref: Reference group; BMI: Body Mass Index; N/A: The number with missing data was very small; thus, the confidence interval could not be computed. * The number with missing data was very small; hence the *p*-value could not be computed. ^a^ Data has already been segregated by sex (No values to input).

**Table 5 ijerph-17-06957-t005:** Results of multivariate logistic regression analysis with elevated blood glucose as outcome among the residents of Kajiado County, Kenya (*n* = 593).

Variable	Total (*n* = 593)		Men (*n* = 221)		Women (*n* = 372)	
	AOR (95% CI)	*p*-Value	AOR (95% CI)	*p*-Value	AOR (95% CI)	*p*-Value
Age (Years)						
25–34	1 (Ref.)		1 (Ref.)		1 (Ref.)	
35–54	1.51 (0.86–2.65)	0.15	2.32 (0.82–6.56)	0.11	1.20 (0.60–2.42)	0.61
55–64	3.59 (1.78–7.25)	<0.01	3.94 (1.15–13.48)	0.03	3.96 (1.58–9.95)	<0.01
Education						
Basic	1 (Ref.)		1 (Ref.)		1 (Ref.)	
High school/higher	0.98 (0.58–1.66)	0.95	1.19 (0.50–2.84)	0.70	0.87 (0.43–1.75)	0.69
Occupation						
Unemployed	1 (Ref.)		1 (Ref.)		1 (Ref.)	
Employed	0.43 (0.20–0.91)	0.03	0.41 (0.09–1.87)	0.25	0.58 (0.21–1.57)	0.28
Self-employed	0.57 (0.33–1.00)	0.05	0.91 (0.24–3.38)	0.89	0.50 (0.26–1.00)	0.05
BMI (Kg/m^2^)						
Normal (<25.0)	1 (Ref.)		1 (Ref.)		1 (Ref.)	
Overweight (25.0–29.9)	1.10 (0.59–2.05)	0.76	2.49 (0.89–6.91)	0.08	0.74 (0.33–1.68)	0.48
Obese (30.0 and above)	3.01 (1.67–5.42)	<0.01	5.33 (1.78–16.00)	<0.01	2.43 (1.16–5.06)	0.02
Waist–hip ratio						
Normal (<0.9/men and <0.85/women)	1 (Ref.)		1 (Ref.)		1 (Ref.)	
High (≥0.9/men and ≥0.85/women)	2.52 (1.43–4.43)	<0.01	1.73 (0.65–4.56)	0.27	3.18 (1.52–6.64)	<0.01
Missing data	1.40 (0.24–8.27)	0.71	8.66 (0.80–93.97)	0.08	N/A	*
Blood pressure(mmHg)						
Normal (<120 and <80)	1 (Ref.)		1 (Ref.)		1 (Ref.)	
Pre-hypertension (120–139/80–89)	1.06 (0.57–1.98)	0.84	1.71 (0.42–6.90)	0.45	0.89 (0.42–1.88)	0.76
Hypertension (≥140 and ≥90)	1.52 (0.78–2.95)	0.22	1.70 (0.40–7.26)	0.47	1.52 (0.68–3.38)	0.31
Low density lipoprotein (mg/dl)						
Optimal (<100)	1 (Ref.)		1 (Ref.)		1 (Ref.)	
High (≥130)	1.11 (0.57–2.13)	0.76	0.64 (0.21–1.99)	0.44	1.41 (0.60–3.33)	0.44
Missing data	1.28 (0.52–3.17)	0.59	0.71 (0.07–7.57)	0.77	1.66 (0.60–4.56)	0.33

CI: Confidence Interval; AOR: adjusted odds ratio (all the variables in the table were mutually adjusted for each other in the multivariable analysis). N/A: The number with missing data was very small; thus, the confidence interval could not be computed. * The number with missing data was very small; hence, the *p*-value could not be computed.

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
