# Peer review of "Prevalence and Risk Factors of Elevated Blood Pressure and Elevated Blood Glucose among Residents of Kajiado County, Kenya: A Population-Based Cross-Sectional Survey"

_ijerph, 2020, doi:10.3390/ijerph17196957_

Round 1
Reviewer 1 Report
The manuscript named “ Prevalence and risk factors of elevated blood pressure 2 and elevated blood glucose among residents of 3 Kajiado County Kenya: A population-based 4 cross-sectional survey” described the status and risk factors of the important diseases of non-communicable diseases (NCDs) and showed some important information which is meaningful in prevention of NCDs in Kenya. In general, this manuscript is fluent and logic, methodological description is clear and the results and discussion are clearly and completely shown.
There are some needed to modify:
Background
- Please provide more characteristics of urbanising area bordering the capital of Kenya in the background.
- Page 2 line 65, please move poor to in front of urban.
Materials and methods
- During the sampling size be calculated, why consider such a higher prevalence of elevated BP (70%)?
- page 4 line 139, please add “.” behind of “measuring tape”.
- page 4 line 148, please add “,” between “previous night” “their appointments”.
4.page 12 line 258, Women were less likely to be physically active than men? Please check this with the table.
Discussion
- page 21 line 322-323, please give the number of the reference in “The national 321 stepwise survey found a similar pattern whereby 56% and 88% of the respondents had never been 322 measured blood pressure and blood glucose, respectively”.
- page 23 line 438,please modify “bias, Moreover, although”.
Author Response
Background
1. Please provide more characteristics of urbanising area bordering the capital of Kenya in the background.
Response
Thank you for this suggestion. We have included additional information regarding the characteristics of urbanizing area bordering the capital of Kenya. It can be read in the manuscript from line 73 to line 86 on page 2.
2. Page 2 line 65, please move poor to in front of urban.
Response
We have fixed this error to read “poor urban (slums) or rural areas” (Page 2, line 66).
Materials and methods
1. During the sampling size be calculated, why consider such a higher prevalence of elevated BP (70%)?
Response:
Thank you for this question. Our estimate was based on literature review. A previous study (Joshi et al.(reference [25] in the manuscript) conducted among adults in an urban slum in Nairobi, found that 1178 (59.3%) adults had pre-hypertension and 205 (10.3%) had hypertension. The prevalence of pre-hypertension tends to be much higher than that of hypertension. Elevated blood pressure consists of both pre-hypertension (SBP 120-139 or DBP 80-89) and hypertension (SBP ≥140 and/or DBP ≥90). The national prevalence of hypertension (SBP ≥ 140 mmHg and/or DBP ≥ 90 mmHg and/or self-report of previous diagnosis) based on the 2015 WHO stepwise survey was 24.5%. This is close to the prevalence of hypertension in our study area (26.2%). The prevalence of pre-hypertension
was 46.5%. The prevalence of elevated Blood Pressure in our study area was 72.7% (Table 2), which is close to the 70% prevalence we used in calculating the required sample size.
2. page 4 line 139, please add “.” behind of “measuring tape”.
Response:
We have added the full stop. (It is now line 153 on page 4)
3. Page 4 line 148, please add “,” between “previous night” “their appointments”.
Response:
We have added the comma. (It is now line 169 on page 4)
4 Page 12 line 258, Women were less likely to be physically active than men? Please check this with the table.
Response:
We have confirmed that this accurate, as presented in Table 3. Nonetheless, we have revised the statement and it now reads as follows: (It is now line 281 on page 11)
“The prevalence of adequate physical activity was 39.5%; being higher in men (45.0%) than in women (33.2%).”
Discussion
1. page 21 line 322-323, please give the number of the reference in “The national stepwise survey found a similar pattern whereby 56% and 88% of the respondents had never been measured blood pressure and blood glucose, respectively”.
Response.
Thank you for this comment.
We have given the number of the stepwise survey. (It is now page 20 line 348). The reference is [49] in the manuscript.
2. page 23 line 438,please modify “bias, Moreover, although”.
Response: We have revised the statement to read:
“Firstly, data on fruit and vegetable consumption, cooking oil and cooking fat use, salt use, tobacco use, alcohol use and physical activity were self-reported and hence prone to information bias. These factors are known to be associated with elevated blood pressure or elevated blood glucose, however, we did not replicate such associations in this study probably due to this bias.” (page 23 line 490 to line 494)
Reviewer 2 Report
This study described the prevalence and risk factors of elevated blood pressure and elevated blood glucose among residents of Kajiado County Kenya based on a cross-sectional survey.
In this paper, the author used the sentence "the prevalence of blood pressure, blood glucose", which is not correct.
Data were collected from Nov. 2016 to Aug 2017, which was too long for a cross-sectional study.
The sample size was too small (n=593) and the proportion of subjects with missing data was too high (20.9%). This would greatly limit the value of this study.
Line 225-226."Men were more likely to be aged 55 -64 years (26.2%) than were women (11.2%). " This description is irrational. The age difference was attributed to the sampling, rather than sex.
Table 1-3. The author only described the difference between men and women, without statistical test.
Table 3 -5. When analyzing AOR, which factors have been adjusted for?
Is it rational to analyze the risk factors based on a cross-sectional survey?
Author Response
In this paper, the author used the sentence "the prevalence of blood pressure, blood glucose", which is not correct.
Response Thank you for pointing this out. We have edited the manuscript so that it reads “prevalence of elevated blood pressure and prevalence of elevated blood glucose.
Data were collected from Nov. 2016 to Aug 2017, which was too long for a cross-sectional study.
Response:
We agree that this was a long period of time for a cross-sectional study. This was due to a number of practical reasons. First, Kajiado county is an expansive area which is primarily semi-arid. The inhabitants are sparsely populated with limited transport network. A vast area is covered by a forest with wild animals and one can only go around to the other side. Secondly, we had to take a break between December and January due to the christmas and new year festivities. Around this time many participants travelled and for those who did not travel, their eating patterns might have changed due to the festivities and this would have affected the results. Therefore, we had to take a break during this time. Thirdly, every enumeration area required two to three days to interview and conduct measurements. For these reasons, a lot of time was spent until the completion of the data collection.
Having said that, it is not unusual for surveys to be conducted over a period of months. For example, the Kenya Demographic and Health Survey (DHS) was conducted from May 7 to October 20, 2014. DHS surveys in some countries have taken longer than this.
Although this study occurred over a long period of time, it is unlikely that the prevalence of blood elevated pressure and elevated blood glucose in the population would change over this period. Therefore, it is our humble view that indeed we estimated the prevalence of elevated blood pressure and elevated blood glucose and its determinants in this study.
The sample size was too small (n=593) and the proportion of subjects with missing data was too high (20.9%). This would greatly limit the value of this study.
Response:
Thank you for this comment. In our sample calculation, we allowed for 90% response rate. We obtained 100% response rate for the questionnaire interviews, however, we could not include in the analysis data for the participants who had incomplete data on biological measurements yielding a 79.1% response rate. The missing data mainly resulted from taking the measurements on the next day after the interview, when the respondents were in a fasting state. When we compared the characteristics of the participants with complete data and those excluded because of incomplete data, we observed no statistically significant difference. This means that those with incomplete data were a random sub-sample of those with complete data, thus, their exclusion is unlikely to have affected our results.
We have included this among limitations of the study (page 23 line 494 to 497).
Line 225-226."Men were more likely to be aged 55 -64 years (26.2%) than were women (11.2%). " This description is irrational. The age difference was attributed to the sampling, rather than sex.
Response:
We agree that this difference may be attributed to the sampling rather than the sex distribution in Kajiado. The sex distribution in Kajiado constitute of 50.2 percent male and 49.8 percent female. We have dropped this statement.
Table 1-3. The author only described the difference between men and women, without statistical test.
Response:
Thank you for this comment. The purpose of Table 1 is to describe the sociodemographic characteristics of the study subjects; total and stratified by sex. Since this table is only descriptive, we have decided to use only descriptive statistics (frequency and percent) in line with STROBES guidelines. In Tables 2-3, we aimed show the prevalence of the risk factors. In these tables, we have included both descriptive and inferential statistics (95% CIs and Ò³2-test P values for comparison between men and women).
Table 3 -5. When analyzing AOR, which factors have been adjusted for? Is it rational to analyze the risk factors based on a cross-sectional survey?
Response:
Thank you for this question.
In analyzing AOR, the variables in the model were mutually adjusted for each other. We have added this in the footnotes of tables 4 and 5. “all the variables presented were mutually adjusted for each other in the multivariable analysis”
It is rational to analyze risk factors using a cross-sectional study, and there are several published studies that have done this (for example https://doi.org/10.1093/ije/dyq156, https://doi.org/10.1093/ije/dyt184). Nonetheless, the use of a cross-sectional design has some limitations, which we have acknowledged in the Discussion section of the manuscript. (page 23 line 504 to line 508). It reads as follows; “Finally, data from a cross-sectional study design prohibits causal interpretation. Our findings may have been affected by reverse causality, where participants with diabetes or hypertension prior to the study might have modified their diet and lifestyle. To mitigate this study design limitation, we recommend setting up large-scale cohort studies to investigate NCD risk factors in Kenya.”
Reviewer 3 Report
The manuscript reviewed entitled “Prevalence and risk factors of elevated blood pressure and elevated blood glucose among residents of Kajiado County Kenya: A population-based cross-sectional survey” describes the rapid increase in risk factors for NCDs with focus on diabetes, obesity and hypertension.
According to the authors this is the first study in Kenya that investigates the urbanizing local
community in terms of NCD risk factors and incorporating the measurement of blood pressure and blood glucose, plasma lipids and urinary Na/K ratio
The manuscript is wellwritten and easy to follow and the Materials and methods part provide good information on the study and available data. It was a pleasure reading it – so well structured and logic.
In the Results section I do not feel this is something that should be separately mentioned in the text: Men were more likely to be aged 55–64 years (26.2%) than were women (11.2%).
Line 240. …were similar ….
Line 252: was the measurement of WHR described in the methods?
I do not think total cholesterol should be included in Table 4 and 5.
Not sure why this sentence is here, could/should be deleted: The fact that the prevalence of overweight and obesity was high even among normal blood pressure and blood glucose groups, particularly in women, may mean that overweight and obesity may be a complex entity that includes at least 2 types that have different degrees of biological relationship with blood pressure and blood glucose.
Line 397 …some word missing ? …less than …..
Author Response
Results: I do not feel this is something that should be separately mentioned in the text: Men were more likely to be aged 55–64 years (26.2%) than were women (11.2%).
Response:
Thank you for pointing out this. We realize that this difference may be attributed to the sampling rather than the sex distribution in Kajiado. Even though there was this difference, we agree that it is not worth mentioning separately in the text. The sex distribution in Kajiado constitute of 50.2 percent male and 49.8 percent female. We have dropped this statement.
Line 240. …were similar ….
Response:
Thank you for this comment.
We have reframed this to read “The prevalence of both pre-hypertension and hypertension was higher in men (49.0% and 31.6%, respectively) than in women (43.7% and 20.1%, respectively), while that of pre-diabetes and diabetes were similar between the sexes (8.5% and 8.0%, respectively in men and 11.6% and 7.4% in women).”
(It is now line 261 on page 8).
Line 252: was the measurement of WHR described in the methods?
Response:
Thank you for pointing this out.
We have described this on (page 4 line 153 to line 159) and it reads as follows;
“Waist circumference was measured over a layer of light clothing. Participants were asked to stand with their feet fairly close together making sure their weight was equally distributed on each leg and were breathing normally. Standing to the side of the participant, the last palpable rib and the top of the hip bone were located. The measuring tape was wrapped around the body in a horizontal position at the midpoint of the last palpable rib and the top of the hip bone making sure the tape was snug but not tight enough to cause compression of the skin. Hip circumference was measured as the maximal circumference over the buttocks.”
I do not think total cholesterol should be included in Table 4 and 5.
Response:
Thank you for this comment.
We agree and have re-analyzed the data and dropped total cholesterol from both tables 4 and 5. This is because this variable is likely to be a mediator.
Not sure why this sentence is here, could/should be deleted: The fact that the prevalence of overweight and obesity was high even among normal blood pressure and blood glucose groups, particularly in women, may mean that overweight and obesity may be a complex entity that includes at least 2 types that have different degrees of biological relationship with blood pressure and blood glucose.
Response:
Thank you for pointing this out.
We agree and have deleted the sentence.
Line 397 …some word missing? …less than ….
Response:
We have revised the sentence to read: “Regarding the relationship between blood pressure and BMI, another interesting finding from our study is that 52% of participants with normal BMI had pre-hypertension.”(It is now on page 22, line 420 to line 421).
Reviewer 4 Report
In this study, the authors sought to evaluate the prevalence of non-communicable disease (NCD) risk factors in an urbanizing Kenyan population. This was a cross-sectional study and they report that a large proportion of the study participants had elevated blood pressure or elevated blood sugar, both NCD risk factors. They also report that there was a high prevalence of unhealthy behaviors, such as low physical activity and low consumption of fruits/vegetables, that are also associated with increased risk of NCD.
Comments:
-Please state the aim of the study in the introduction.
-The supplementary tables need to be referenced within the text.
-The aim of the study was to gather data to inform interventions to mitigate the risks of NCD. The authors should elaborate in the discussion sections what these interventions should entail in their studied population.
Author Response
Please state the aim of the study in the introduction.
Response:
Thank you for pointing out this.
We have included the aim of the study in the introduction (Page 2 line 68 to line 70):
“Thus, we aimed to determine the prevalence of elevated blood pressure, elevated blood glucose and their determinants in Kajiado county, a typical rapidly urbanising area bordering the capital of Kenya.”
The supplementary tables need to be referenced within the text.
Response:
Thank you for this comment.
We have referenced supplementary tables in the text as tables S1 – S4.
The aim of the study was to gather data to inform interventions to mitigate the risks of NCD. The authors should elaborate in the discussion sections what these interventions should entail in their studied population.
Response
Thank you for this comment.
We have elaborated in the discussion section the details that should be included in the interventions in our study area. This can be found on page 22, line 447 to page 23, line 476.